cognition/psychology/behaviour

paranoia, social inference, psychosis, interpersonal sensitivity

**Author for correspondence:**
J. M. Barnby
e-mail: joe.barnby@kcl.ac.uk

†These authors contributed equally to the work.

# Paranoia, sensitization and social inference: findings from two large-scale, multi-round behavioural experiments

J. M. Barnby[1], Q. Deeley[2], O. Robinson[3], N. Raihani[4], V. Bell[1,5,†] and M. A. Mehta[1,†]

[1]Social and Cultural Neuroscience Research Group, Centre for Neuroimaging Sciences, Institute of Psychiatry, Psychology, and Neuroscience, and [2]Social and Cultural Neuroscience Research Group, Forensic and Neurodevelopmental Sciences, Institute of Psychiatry, Psychology, and Neuroscience, King's College London, London, UK
[3]Institute of Cognitive Neuroscience, [4]Psychology and Language Sciences, and [5]Research Department of Clinical, Educational, and Healthy Psychology, University College London, London, UK

JMB, 0000-0001-6002-1362

The sensitization model suggests that paranoia is explained by over-sensitivity to social threat. However, this has been difficult to test experimentally. We report two preregistered social interaction studies that tested (i) whether paranoia predicted overall attribution and peak attribution of harmful intent and (ii) whether anxiety, interpersonal sensitivity and worry predicted the attribution of harmful intent. In Study 1, we recruited a large general population sample ($N = 987$) who serially interacted with other participants in multi-round dictator games and matched to fair, partially fair or unfair partners. Participants rated attributions of harmful intent and self-interest after each interaction. In Study 2 ($N = 1011$), a new sample of participants completed the same procedure and additionally completed measures of anxiety, worry and interpersonal sensitivity. As predicted, prior paranoid ideation was associated with higher and faster overall harmful intent attributions, whereas attributions of self-interest were unaffected, supporting the sensitization model. Contrary to predictions, neither worry, interpersonal sensitivity nor anxiety was associated with harmful intent attributions. In a third exploratory internal meta-analysis, we combined datasets to examine the effect of paranoia on trial-by-trial attributional changes when playing fair and unfair dictators. Paranoia was associated with a greater reduction in harmful intent attributions when playing a fair but not unfair

dictator, suggesting that paranoia may also exaggerate the volatility of beliefs about the harmful intent of others.

# 1. Background

Paranoia is a common feature in psychosis and involves an unfounded belief that others intend harm, now or in the future [1]. Paranoid beliefs can be induced by recreational drugs [2,3], following sleep deprivation [4] during or after seizures [5], or from being subject to high stress [6]. Paranoia also exists as a continuous trait in the general population and has shown to be characterized by interpersonal sensitivity, mistrust, ideas of reference and ideas of persecution [7,8].

Once developed, paranoid beliefs are maintained by several personal and interpersonal factors. On the personal level, worry, insomnia [9], anxiety [10,11], probabilistic reasoning biases [12], belief inflexibility [13] and safety behaviours (avoiding the source of perceived threat) [11] all contribute to paranoia. Interpersonal cognitive biases also affect how individuals interpret social situations. The most established effect is that those with paranoid beliefs have an externalizing attribution bias, whereby causes of negative events are more likely to be attributed to other people [14]. Trait interpersonal sensitivity has also been associated with paranoid thinking. Those at high risk of developing psychosis report increased paranoid thinking following simulated interactions in a virtual social environment which was predicted [15] or mediated [16] by interpersonal sensitivity.

The sensitization model of psychosis argues that environmental stresses and genetic vulnerabilities sensitize biological, cognitive and affective processes to produce symptoms of psychosis, and importantly, paranoid beliefs [17–19]. Neuroimaging studies have observed increased presynaptic dopamine leading up to [20] and during [21] the development of psychotic symptoms, suggesting aberrant dopaminergic transmission as crucial in sensitization [22]. Experimental data support the sensitization of cognitive and affective processes that manifests as a 'jumping to conclusions' probabilistic reasoning bias [12,23], high initial mistrust [24,25] and more threatening or negatively valanced responses following heightened social arousal [26,27].

One prediction arising from this model is that those high in paranoid ideation will show increased sensitivity to interpersonal interactions, and specifically potential or actual social threat, leading to an increased tendency to attribute harmful intent to others, putatively both more quickly and to a greater degree.

Economic games derived from game theory have been previously used to test the effect of paranoia on intention attributions. These games allow for social interactions within a tightly controlled environment. Participants make decisions that have outcomes with genuine gains and losses and therefore real, albeit small, harms and benefits [24,28]. Existing research has shown that increases in harmful intent attributions are associated with trait paranoia, social threat [28,29], social cohesion of partners in a game [30] and greater relative social rank, and outgroup status, of the interaction partner [31]. However, current game theory paradigms in paranoia research that have allowed for participant-to-participant (rather than simulated; [15,16]) interactions have tended to use single round games or brief interactions that are not able to test the effect of paranoia and additional psychological variables on attributions over evolving interactions.

In this study, we implemented a multiple-round game theory interaction using serial dictator games. The dictator game has widely been used in paranoia research [28,29,31] and involves a situation where two participants are paired and one (the 'dictator') is given a sum of money that they can choose to share with the 'receiver' participant [32]. The receiver has no control and must accept any amount that the dictator offers. The game has been previously modified to assess social inferences made by the receiver [28,29]. After each interaction, receivers are required to rate to what extent the dictators were motivated by self-interest or an intent to harm. In the paradigm developed for this study, participants completed six serial dictator trials against fair, partially fair and unfair partners, while rating harmful intent and self-interest motivating their partner's actions, allowing a test of sensitivity over evolving social interactions. This also allowed us to test the effect of several key affective processes previously identified as important in paranoia, namely anxiety, worry and interpersonal sensitivity.

The sensitization model of paranoia suggests several hypotheses we tested over two studies. In Study 1, we hypothesized that high levels of paranoid ideation would predict earlier and larger harmful intent attributions during the multi-round interaction. In Study 2, we hypothesized that harmful intent attributions would be predicted by anxiety, interpersonal sensitivity and worry. Studies 1 and 2 were preregistered and included hypotheses designed to replicate findings from previous studies (high

attribution of harmful intent is associated with higher paranoia and unfair dictators; [28–31]) as well as the key experimental hypotheses described above. Finally, we combined data from Studies 1 and 2 to complete exploratory analysis to gain better resolution on trial-by-trial effects, dictator exposure effects and dictator behaviour overall.

# 2. Study 1

This study tested the main hypothesis that paranoid ideation predicts in-the-moment harmful intent attributions within serial interpersonal interactions, both in terms of overall value and by how quickly individuals reach a marker of high harmful intent attribution. Specifically, in line with prior work [28,29], we predicted that pre-existing paranoia and more unfair dictator behaviour would lead to higher harmful intent attributions. In line with prior theory [26,27] suggesting initial sensitization following negatively valenced responses, we also predicted that paranoia would lead to fewer trials before a high harmful intent score was reached (specific preregistered predictions for this study can be found here https://aspredicted.org/ka4ny.pdf).

## 2.1. Methodology

This project was approved by the King's College London ethics board (Study 1: MRS-17/18-8312). All data were collected in September 2018 using Prolific Academic (hereafter Prolific; www.prolific.ac), an online crowd-sourcing platform. All data and analysis scripts are available online (https://osf.io/u92rg/).

Prior to taking part in both studies, participants were informed that their participation was voluntary, and were required to tick a box giving consent for the authors to use their anonymous data for research purposes. Using Prolific allowed the rapid recruitment of a more demographically diverse sample of participants than recruitment from our social media or university networks [33]. We included participants from the UK who were fluent in English and had no current or history of mental illness.

We recruited 987 participants (372 males). Of these, 226 people would be required to detect an effect size of 0.1 with at least seven predictors in a multiple regression model. In order to produce robust inferences, we recruited the maximum number of participants that our resources would allow. Participants first completed the Green Paranoid Thoughts Scale (GPTS; [34]). Participants were asked to indicate the extent of feelings described in 32 statements using a Likert scale of 1–5, where 1 = not at all and 5 = totally. Scores can range from 32 to 160, with higher scores indicating a greater degree of paranoia. The GPTS was chosen as a suitable measure as it includes both core aspects of the definition of paranoia (1): social concerns about others and perception of intended harm. It has also shown to be the most reliable and valid scale for measuring paranoia across the clinical and non-clinical spectrum [35]. Total paranoia scores were obtained for each participant by summing the response scores to all questions, comprising both the social reference and the persecution scales. Hereafter, this variable is referred to as 'paranoia'.

After completing the survey, and in keeping with Raihani & Bell [28,29], we allowed a minimum interval of 7 days to elapse before inviting all participants to take part in the multi-round dictator game.

We developed a within-subjects, multi-trial modification of the dictator game design used in previous studies to assess paranoia (see electronic supplementary material, appendix A; [28,29]). Each participant played six trials against three different types of dictator. In each trial, participants were told that they had been endowed with a total of £0.10 and their partner (the dictator) had the choice to take half (£0.05) or all (£0.10) the money from the participant. Dictators were set to either always take half of the money, have a 50:50 chance to take half or all of the money or always take all of the money. This was noted in this study as fair, partially fair and unfair, respectively. The order that participants were matched with dictators was randomized. Each dictator had a corresponding cartoon avatar with a neutral expression to support the perception that each of the six trials was with the same partner.

After each trial, participants were asked to rate on a scale of 1–100 (initialized at 50) to what degree they believed that the dictator was motivated (i) by a desire to earn more (self-interest) and (ii) by a desire to reduce their bonus in the trial (harmful intent). Following each block of six trials, participants were asked to rate the character of the dictator overall by scoring intention again on both scales. Therefore, participants judged their perceived intention of the dictator on both a trial-by-trial and summary level.

After making all 42 attributions (two attributions for each of the six trials over three partners, plus three additional overall attributions for each partner), participants were put in the role of the dictator for six trials—whether to make a fair or unfair split of £0.10. Participants were first asked to choose

an avatar from nine different cartoon faces before deciding on their six different splits. These dictator decisions were primarily collected to truthfully inform participants that decisions were made by real people (as in prior studies using this method, see [28,29]). We also included the decisions made by participants in an exploratory analysis.

This modification to the original dictator game design allowed us to track how changes in pre-existing paranoia where associated with changes in attributions about partner behaviour and the order of initial partner exposure and whether attributions were highly variable over trials or consistent. Eight hundred and twelve participants (294 males) were able to be followed up to play the multi-round dictator game. The mean age range of participants was 36–40 in the second sample.

All participants were paid for their completion of the GPTS, regardless of follow-up. Participants were paid a baseline payment for their completion of the dictator game, along with any additional bonuses won in the game.

### 2.1.1. Analysis

Analyses conform to those outlined in our preregistration unless stated otherwise.

This study used an information-theoretic approach for confirmatory analysis. We analysed the data using multi-model selection with model averaging (described in [28,29]). The Akaike information criterion, corrected for small sample sizes (AICc), was used to evaluate models, with lower AICc values indicating a better fit [36]. The best models are those with the lowest AICc value. To adjust for the intrinsic uncertainty over which model is the true 'best' model, we averaged over the models in the top model set to generate model-averaged effect sizes and confidence intervals [37]. In addition, parameter estimates and confidence intervals are provided with the full global model to robustly report a variable's effect in a model [38]. This used package 'MuMIn' (v. 1.43.1; [39]). All analyses were performed in R (v. 3.6.0; [40]) on an Apple OSX operating system (Mojave, 10.14.6). Visualizations were generated using the package 'ggplot2' (v. 3.2.1; [41]).

In our models, baseline continuous scale scores were centred and scaled to produce Z-values. Model statistics reported are β-coefficients.

Average scores of harmful intention attributions and self-interest for each dictator were taken over each six trials for trial analysis. These were used for cumulative link mixed-models (clmm; [42]). Harmful intent and self-interest attributions were set as our dependent variable. Paranoia, dictator order, dictator behaviour (fair, unfair and partially fair), age, sex and paranoia × dictator behaviour were set as our explanatory terms with ID set as the random term.

For our third prediction, participants that scored above 60 were considered to have scored high harmful intent attributions. In both harmful intent and self-interest scores, participants were set a value of 6 if they had scored 60 in their first trial, 5 if they had scored over 60 by their second trial, 4 if they had scored 60 by their third trial and so on. We report this result, but also wanted to consider a high harm attribution as someone that scored over the mean harmful intent attribution of the population for each dictator. This is also reported in addition to our preregistered plan, which was based on previous mean group estimates. Mean thresholds for each dictator are stated for each analysis in the 'Results' section. All trials following the threshold being reached were coded as 0. Participants not reaching the threshold for any trial were coded 0 across all trials. Both unfair and fair dictator behaviour were analysed with two cumulative link models (clm) each, one for harm-intent and one for self-interest. This slightly deviates from our preregistration that suggests the use of Kruskal–Wallis and Dunn *post hoc* tests; however, we decided that using a clm is a more robust way to analyse the data.

For visualization purposes, we calculated paranoia groups based on the quantiles of GPTS scores across the population and additionally divided those in the top quantile by those exceeding the clinical mean of paranoia defined in previous work (101.9; [35]). These divisions were: low (less than 36; $n = 232$), medium (36–43; $n = 180$), high (44–59; $n = 199$), very high (59–101.9; $n = 167$) and clinical (greater than 102, $n = 34$). This variable is hereafter named paranoia 'level'. Slightly different score parameters for each paranoia level were included in our preregistration, but we have adapted them in this study based on our population GPTS quartiles.

## 2.2. Results

Eight hundred and twelve participants that were able to be followed up were included in the analysis. Fifteen were removed for incomplete data, 24 removed for failing both control questions and 136 for

**Table 1.** Variables affecting harmful intention and self-interest scores in the multi-round dictator game (Study 1). Harmful intent was coded as a five-level ordinal categorical variable and set as the response term in the clmm. ID was set as the random variable [42]. Relative importance is the probability that the term in question is a component of the true best model and a value for the amount of times the term is included in the selection of top models to be averaged. Order refers to the order in which a fair, partially fair or unfair dictator was presented to participants. An interaction between dictator and paranoia is not included in the model for self-interest attributions as it was not included in the final top model. Age was not included in the final top model and is therefore absent from the tables.

| parameter | estimate | standard error | 95% CI lower | upper | relative importance |
|---|---|---|---|---|---|
| harmful intent attributions | | | | | |
| Intercept 1\|2 | −1.26 | 0.11 | −1.48 | −1.05 | |
| Intercept 2\|3 | 0.47 | 0.10 | 0.27 | 0.68 | |
| Intercept 3\|4 | 2.17 | 0.12 | 1.94 | 2.39 | |
| Intercept 4\|5 | 3.67 | 0.14 | 3.41 | 3.94 | |
| dictator | | | | | |
| (fair < partially fair < unfair) | 2.22 | 0.09 | 2.06 | 2.39 | 1 |
| order | | | | | |
| (fair < partially fair < unfair) | −1.12 | 0.15 | −1.42 | −0.83 | 1 |
| paranoia (Z-score) | 0.36 | 0.09 | 0.19 | 0.53 | 1 |
| sex (male \| female) | −0.03 | 0.11 | −0.26 | 0.19 | 0.25 |
| dictator × paranoia | 0.14 | 0.10 | −0.06 | 0.34 | 0.79 |
| self-interest attributions | | | | | |
| Intercept 1\|2 | −6.53 | 0.25 | −7.01 | −6.05 | |
| Intercept 2\|3 | −5.25 | 0.21 | −5.66 | −4.84 | |
| Intercept 3\|4 | −3.15 | 0.16 | −3.46 | −2.84 | |
| Intercept 4\|5 | −0.28 | 0.11 | −0.50 | −0.07 | |
| dictator | | | | | |
| (fair < partially fair < unfair) | 4.33 | 0.17 | 3.99 | 4.67 | 1 |
| order | | | | | |
| (fair < partially fair < unfair) | −0.82 | 0.16 | −1.13 | −0.50 | 1 |
| paranoia (Z-score) | 0.01 | 0.05 | −0.09 | 0.11 | 0.24 |
| sex (male \| female) | −0.03 | 0.11 | −0.23 | 0.18 | 0.23 |

non-participation in the multi-round dictator game. Mean baseline paranoid ideation in the excluded participants (mean = 50.43, s.e. = 1.62, range = 32–134) was comparable to those that were included in the analysis ($t_{252} = 0.322$, 95% CI: −2.93, 4.08).

## 2.2.1. Explanatory variables of baseline paranoia score

Paranoia scores ranged from 32.0 to 149.0, with a mean of 51.0 (s.e.: 0.74; skew: 1.7). Older participants were less paranoid (−1.89; 95% CI: −2.22, −1.57), male participants were more paranoid (0.17; 95% CI: 0.04, 0.34) and there was no effect of education on paranoia (−0.39; 95% CI: −1.16, 0.17).

## 2.2.2. Prediction 1: paranoia and harmful intent

As predicted, paranoia positively predicted higher harmful intent (HI) attributions across all three dictators; however, there was no effect of paranoia on self-interest (SI) attributions (table 1).

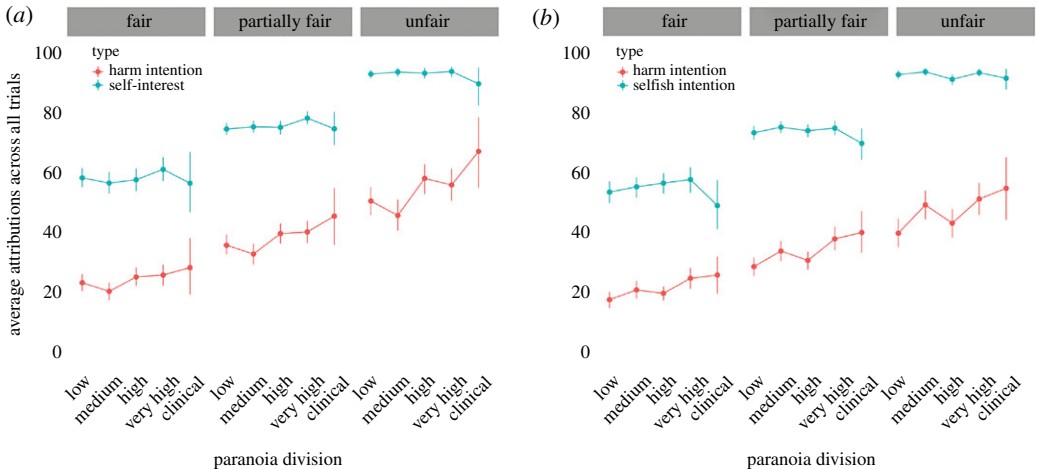

**Figure 1.** Average self-interest attributions and harmful intent attributions, averaged across trials for divisions of GPTS score and faceted by each type of dictator for both Study 1 (*a*; *n* = 812) and Study 2 (*b*; *n* = 885). Dots represent the mean for each level of paranoia. Lines represent the 95% confidence interval. Participants played against different partners in a pseudo-random order. 'Clinical' refers to participants in the general population who scored past the threshold for GPTS scores typical in clinical populations (101.9; [34]).

### 2.2.3. Prediction 2: dictator behaviour and harmful intent

As predicted, as dictators were increasingly unfair (higher proportion of unfair decisions), higher HI and SI attributions were observed (table 1). Figure 1*a* depicts the difference in HI and SI attributions between the population when delineated by their paranoia level (low, medium, high, very high and clinical) for Study 1.

### 2.2.4. Prediction 3: paranoia and earlier high harmful intent attributions

As predicted, high (over 60) harmful intent attributions were triggered in earlier trials as paranoia increases for both unfair (−0.12; 95% CI: −0.21, −0.03) and fair (−0.14, 95% CI: −0.33, −0.01) dictators; however, this was not found for SI attributions (see electronic supplementary material, appendix B).

### 2.2.5. Exploratory analysis

We also completed an analysis using a relative threshold for earlier high decisions based on the mean of the population for each dictator, rather than a pre-set cut-off of 60 as in the preregistered analysis. For unfair dictators, high (mean = 53.51) HI attributions were triggered in earlier trials as paranoia increased (−0.12; 95% CI: −0.20, −0.02). However, this was not found for fair dictators (mean = 24.26) (−0.06; 95% CI: −0.19, 0.01). This was not found for SI attributions in either dictator condition (see Figure 2*a* for trial-by-trial average attributions across participants for Study 1).

## 3. Study 2

Study 1 suggested that prior paranoid beliefs led to larger and earlier harmful intent attributions. Prior models and evidence [9–11,15,16] suggest a role of affective processes in paranoid ideation, specifically interpersonal sensitivity [43], state and trait anxiety [44] and worry [45]. Therefore, we also wanted to test whether these psychological variables predict harmful intent attributions. Specifically, we predicted that scores on measures of state anxiety and overall interpersonal sensitivity (and specifically subscales of 'fragile inner-self' and 'interpersonal awareness') would be associated with higher harmful intent attributions. We also predicted that state anxiety and pre-existing paranoia would interact and be associated with higher and earlier scores of harmful intent attributions (specific preregistered predictions for this study can be found in http://aspredicted.org/yz5gr.pdf).

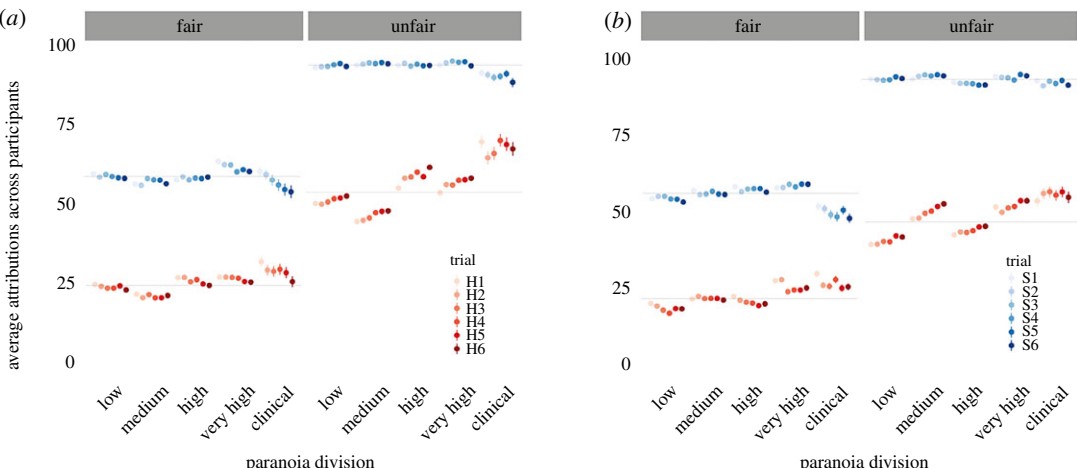

**Figure 2.** Average harmful intent (H1–H6) and average self-interest (S1–S6) attributions for each trial across divisions of GPTS scores, faceted by the type of dictator for Study 1 (*a*; *n* = 812) and Study 2 (*b*; *n* = 885). Points = mean, bars = 95% confidence interval. Grey lines = mean score across the group. 'Clinical' refers to participants in the general population who scored past the threshold for GPTS scores typical in clinical populations (101.9; [34]).

## 3.1. Methodology

This project was approved by the King's College London ethics board (Study 2: LRS-18/19-9281). Data were collected in February 2019 using Prolific. Data and analysis scripts are available online (https://osf.io/u92rg/).

We recruited 1011 participants (374 males). Two hundred and twenty-six people would be required to detect an effect size of 0.1 with at least seven predictors in a multiple regression model. In order to produce robust inferences, we recruited the maximum number of participants that our resources would allow. Participants recruited for this study were not participants in Study 1. Study procedures and analyses were identical to Study 1 aside from the inclusion of anxiety, worry and interpersonal sensitivity measures.

We assessed both trait anxiety and state anxiety using the State-Trait Anxiety Inventory (STAI; [44]). It comprises two subscales, one for trait and one for state anxiety, each made of 20 items. Each item is rated on a scale of 1–4, from 'almost never' to 'almost always'. The trait measure was given to participants at baseline alongside the GPTS. The state measure was given immediately after the multi-round dictator game.

We measured interpersonal sensitivity using the Interpersonal Sensitivity Measure (ISM; [43]). The ISM comprises five subscales: fragile inner-self (five items), need for attachment (eight items), interpersonal awareness (seven items), timidity (eight items) and separation anxiety (eight items). Each item is on a scale of 1–4, from 'very unlike you' to 'very like you'. Subscales are summed to form summary scores. The ISM was given at baseline alongside the GPTS.

We also measured worry using the Penn State Worry Questionnaire (PSWQ) [45] as worry has been additionally implicated as highly predictive of paranoia [1]. The PSWQ comprises 16 items, each on a scale of 1–5, from 'not at all typical of me' to 'very typical of me'. The PSWQ was given at baseline alongside the GPTS.

All analyses were performed in R (v. 3.6.0; [40]) on an Apple OSX operating system (Mojave, 10.14.6).

Analyses conform to our preregistration unless stated otherwise. We included the explanatory variables from the STAI, PSWQ and ISM in our cumulative link mixed models alongside the GPTS scores with the ID set as the random variable. Continuous variables were *z*-score transformed. Model statistics reported are β-coefficients unless stated otherwise.

## 3.2. Results

Eight hundred and eighty-five participants that were able to be followed up were included in the analysis. Eight were removed for incomplete data and 118 for non-participation in the multi-round dictator game. Mean baseline paranoid ideation in the excluded participants (mean = 58.54, s.e. = 2.35, range = 32–140) was higher than participants that were included in the analysis ($t_{153}$ = −2.41, 95% CI: −10.85, −1.09) by a small amount.

**Table 2.** Variables affecting harmful intent and self-interest scores in the multi-round dictator game (Study 2). Harmful intent was coded as a five-level ordinal categorical variable and set as the response term in the clmm. ID was set as the random variable [43]. Relative importance is the probability that the term in question is a component of the true best model and a value for the amount of times the term is included in the selection of top models to be averaged. Order refers to the order in which a fair, partially fair or unfair dictator was presented to participants.

| parameter | estimate | standard error | 95% CI | | relative importance |
| --- | --- | --- | --- | --- | --- |
| | | | lower | upper | |
| *harmful intent attributions* | | | | | |
| Intercept 1\|2 | −0.64 | 0.23 | −1.09 | −0.18 | |
| Intercept 2\|3 | 1.28 | 0.24 | 0.82 | 1.74 | |
| Intercept 3\|4 | 2.95 | 0.25 | 2.47 | 3.43 | |
| Intercept 4\|5 | 4.38 | 0.26 | 3.88 | 4.89 | |
| dictator | | | | | |
| (fair < partially fair < unfair) | 2.00 | 0.09 | 1.82 | 2.18 | 1 |
| order | | | | | |
| (fair < partially fair < unfair) | −1.17 | 0.17 | −1.52 | −0.83 | 1 |
| paranoia (Z-score) | 0.35 | 0.10 | 0.15 | 0.54 | 1 |
| sex (male \| female) | −0.16 | 0.21 | −0.71 | 0.10 | 0.52 |
| age | 0.00 | 0.01 | −0.01 | 0.02 | 0.32 |
| *self-interest attributions* | | | | | |
| Intercept 1\|2 | −6.59 | 0.35 | −7.27 | −5.91 | |
| Intercept 2\|3 | −5.35 | 0.33 | −5.99 | −4.71 | |
| Intercept 3\|4 | −3.16 | 0.30 | −3.75 | −2.58 | |
| Intercept 4\|5 | −0.21 | 0.28 | −0.75 | 0.33 | |
| dictator | | | | | |
| (fair < partially fair < unfair) | 4.59 | 0.17 | 4.26 | 4.93 | 1 |
| order | | | | | |
| (fair < partially fair < unfair) | −0.71 | 0.16 | −1.02 | −0.39 | 1 |
| paranoia (Z-score) | −0.03 | 0.07 | −0.28 | 0.09 | 0.34 |
| sex (male \| female) | 0.01 | 0.07 | −0.31 | 0.43 | 0.11 |
| age | 0.00 | 0.01 | −0.02 | 0.00 | 0.43 |

### 3.2.1. Explanatory variables of baseline paranoia

Paranoia scores ranged from 32 to 159 with a mean of 53 (s.e.: 0.45; skew: 1.54). Older participants were less paranoid (−0.05; 95% CI: −0.05, −0.04), there was a negligible effect of being male on paranoia (0.05; 95% CI: −0.04, 0.24) and there was a quadratic (−1.20, 95% CI: −1.80, −0.60) relationship between education and paranoia. Paranoia positively correlated with anxiety, worry and interpersonal sensitivity ($R = 0.38–0.51$, see electronic supplementary material, appendix C).

### 3.2.2. Replication of main findings of Study 1

Paranoia positively predicted higher HI attributions across all three dictators, there was no effect of paranoia on SI attributions and additionally, unfairness of dictator was associated with higher HI and SI attributions. Order effects were also replicated (see figure 1 and table 2).

For unfair dictators, high (mean = 46.56) HI attributions were not uniformly observed in earlier trials as paranoia increased (−0.06; 95% CI: −0.17, 0.01), but were for fair dictators (mean = 21.39) (−0.12; 95% CI: −0.20, −0.03). However, paranoia was not associated with high SI attributions in earlier trials in either dictator condition.

**Table 3.** Summary of extra explanatory variables affecting harmful intention and self-interest attributions in the multi-round dictator game (Study 2). Harmful intent was coded as a five-level ordinal categorical variable and set as the response term in the clmm. ID was set as the random variable [41]. Relative importance is the probability that the term in question is a component of the true best model. All predictors were run in separate models with dictator, age and sex as other fixed effects. NA, not included in the final top model. Electronic supplementary material, appendix D contains all the estimates of predictors in the models that included paranoia.

| parameter | estimate | standard error | 95% CI lower | upper | relative importance |
|---|---|---|---|---|---|
| *harmful intent attributions* | | | | | |
| trait anxiety | 0.01 | 0.03 | −0.04 | 0.07 | 0.19 |
| state anxiety | 0.04 | 0.06 | −0.09 | 0.18 | 0.37 |
| interpersonal sensitivity | −0.01 | 0.02 | −0.06 | 0.04 | 0.19 |
| interpersonal awareness | −0.13 | 0.11 | −0.36 | 0.09 | 0.70 |
| separation anxiety | 0.08 | 0.14 | −0.13 | 0.28 | 0.51 |
| timidity | −0.11 | 0.12 | −0.34 | 0.12 | 0.63 |
| need for attachment | −0.22 | 0.10 | −0.42 | −0.01 | 1 |
| fragile inner-self | 0.07 | 0.09 | −0.13 | 0.26 | 0.49 |
| worry | 0.06 | 0.10 | −0.13 | 0.24 | 0.42 |
| *self-interest attributions* | | | | | |
| trait anxiety | 0.01 | 0.05 | −0.11 | 0.25 | 0.17 |
| state anxiety | 0.09 | 0.1 | −0.12 | 0.29 | 0.57 |
| interpersonal sensitivity | NA | NA | NA | NA | NA |
| interpersonal awareness | 0.03 | 0.08 | −0.11 | 0.17 | 0.34 |
| separation anxiety | −0.00 | 0.05 | −0.10 | 0.09 | 0.26 |
| timidity | −0.20 | 0.09 | −0.38 | −0.02 | 1 |
| need for attachment | 0.20 | 0.09 | 0.02 | 0.38 | 1 |
| fragile inner-self | −0.03 | 0.06 | −0.15 | 0.10 | 0.35 |
| worry | 0.11 | 0.11 | −0.11 | 0.32 | 0.66 |

Figure 2*b* shows average trial-by-trial attributions for each level of paranoia in Study 2.

### 3.2.3. Predictions 1 and 2: state anxiety, paranoia and harmful intent

Contrary to predictions, state anxiety did not predict overall HI or SI attributions in any dictator condition and there was no interaction with paranoia. Due to the potential collinearity between state anxiety and paranoia, we also ran each model excluding paranoia, although excluding paranoia did not change the conclusions of the estimates (table 3).

### 3.2.4. Predictions 3 and 4: interpersonal sensitivity, paranoia and harmful intent

Contrary to predictions, interpersonal sensitivity predicted a decrease in overall HI (−0.29, 95% CI: −0.49, −0.10) but not SI attributions across all dictators, and there was no interaction between interpersonal sensitivity and paranoia for HI or SI attributions across all dictators. However, excluding paranoia from the models removed the association of interpersonal sensitivity on HI attributions (table 3).

In line with our preregistration, we ran a model with all subscales of interpersonal sensitivity included. This suggested that 'interpersonal awareness' (−0.54, 95% CI: −0.80, −0.28) was negatively associated with HI attributions, while 'separation anxiety' (0.36, 95% CI: 0.08, 0.64) was positively associated. Conversely, 'timidity' was negatively associated with SI attributions (−0.46, 95% CI: −0.68, −0.23), and 'interpersonal awareness' (0.31, 95% CI: 0.04, 0.58) and 'need for attachment' (0.28, 95% CI: 0.07, 0.48) were positively associated. Full model statistics that included all predictors together with paranoia can be found in electronic supplementary material, appendix D (figure 3).

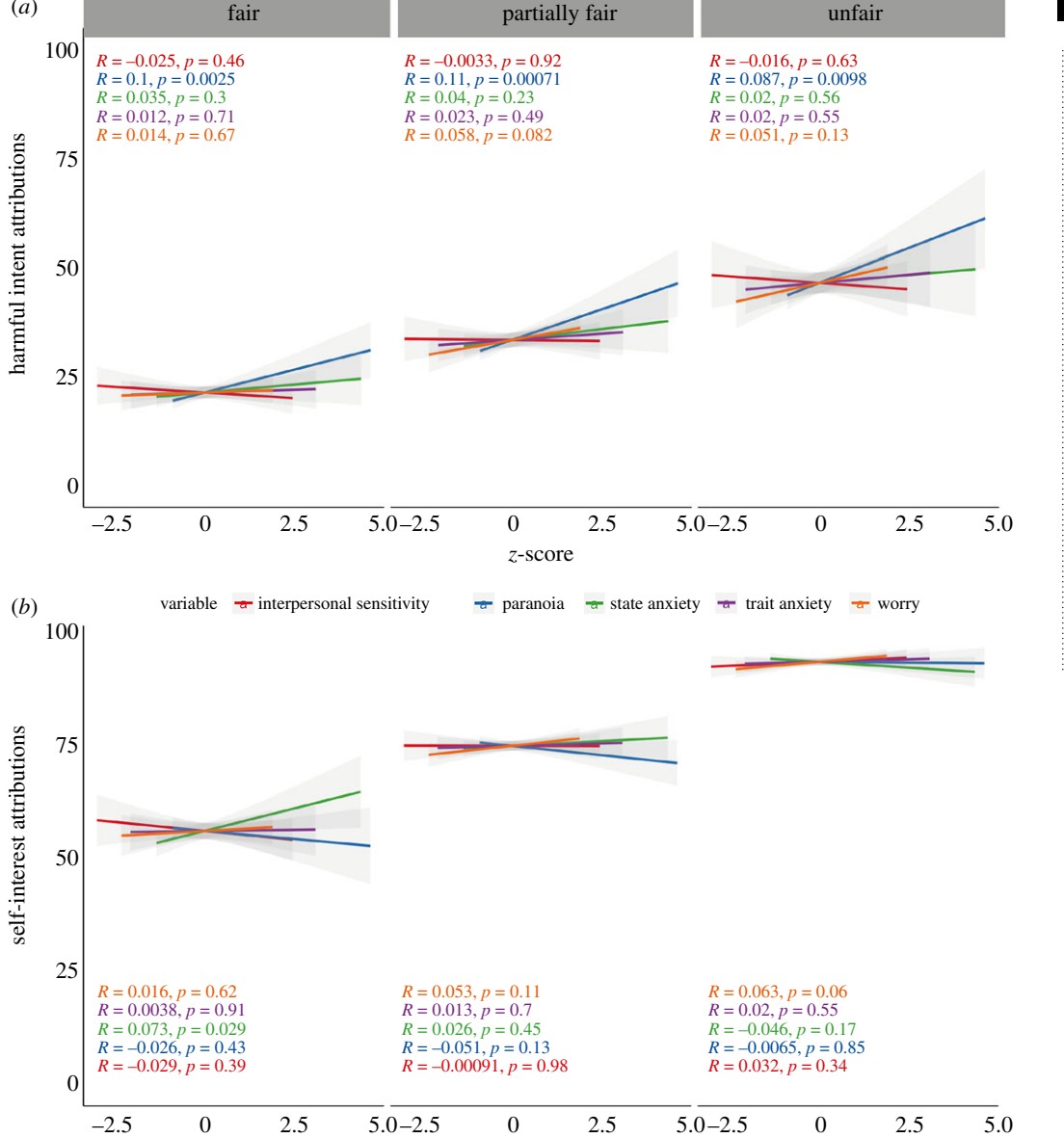

**Figure 3.** Pearson $R$ correlations for centred and scaled scores on state and trait anxiety, paranoia, interpersonal sensitivity and worry questionnaires by harmful intent (*a*) and self-interest (*b*) scores in Study 2, faceted by the dictator condition ($N = 885$).

We also ran all subscales of the interpersonal sensitivity measure in separate models to account for potential collinearity. The 'need for attachment' subscale of the ISM was associated with a decrease in HI attributions, and all other subscales had no effect. 'Timidity' was associated with reduced SI attributions, and 'need for attachment' was associated with an increase in SI scores (table 3).

### 3.2.5. Prediction 5: anxiety, paranoia and trials to peak decision

Contrary to predictions, state anxiety alone and its interaction with paranoia did not predict scoring above the mean in an earlier trial for HI and SI attributions during both unfair and fair dictators. Excluding paranoia from the models did not change the conclusions of the estimates (table 3).

## 4. Internal meta-analysis

We combined data from Studies 1 and 2 to analyse the overall effect of paranoia, trial-by-trial attributional change for each dictator, as well as order effects, and overall dictator behaviour on attributions. We also include an exploratory analysis recommended by a reviewer to assess whether

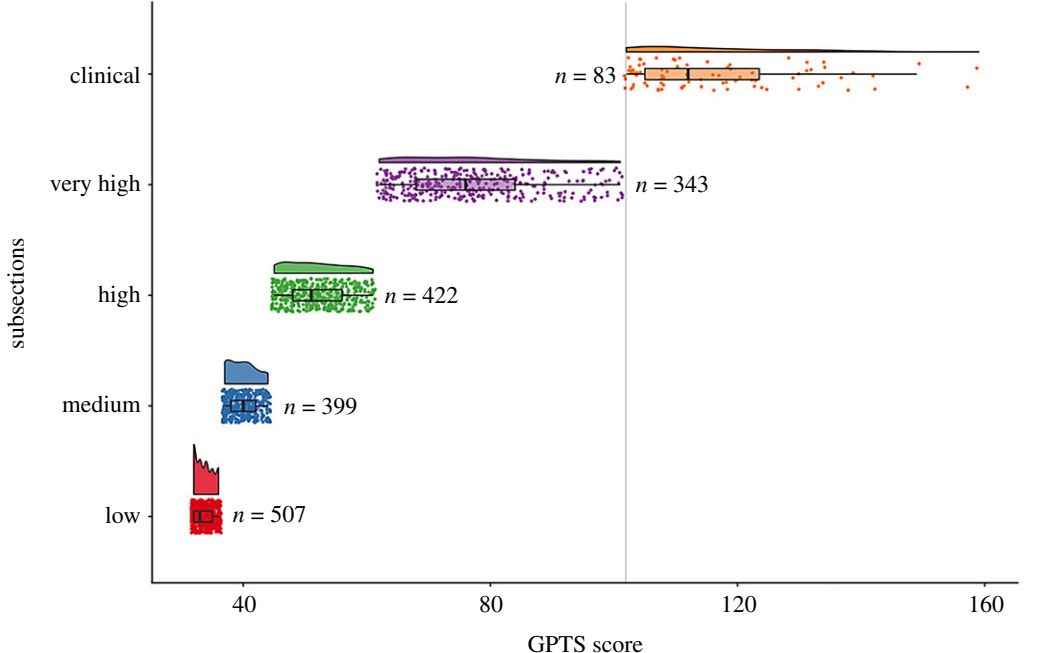

**Figure 4.** Rainbow cloud plot for each quartile of the GPTS. The highest quantile was subdivided into those who had and had not passed the clinical threshold (101.9) [34]. The clinical division is denoted by a grey line.

overall harmful intent and self-interest attributions made across partners in the task, and pre-existing paranoia, affected unfair decisions made by participants when they took the role of the dictator with a new partner.

## 4.1. Methodology

A total of 1754 participants were included in the analysis from Studies 1 and 2. The meta-analysis was not preregistered, although data and analysis scripts are available online (https://osf.io/u92rg/).

As in both previous studies, paranoia scores on the GPTS were divided into quantiles (low, 32–36; medium, 37–44; high, 45–61; very high, 61–101.9) and also a group who passed GPTS scores exceeding the clinical mean (clinical, greater than 101.9) (figure 4).

All analyses were performed in R (v. 3.6.0; [40]) on an Apple OSX operating system (Mojave, 10.14.6).

Linear mixed-effects models (function 'lmer'; package 'lme4'; [46], ID as the random variable) were run to determine the effect of initial dictator exposure on overall HI and SI attributions for fair and unfair dictators. They were also used to calculate changes in HI and SI attributions for each trial relative to the first, and the overall effect of paranoia and sex on attributions. Probability distributions and uncertainty estimates of the direction of β-coefficients produced by mixed effect models were computed for HI and SI attributions for each trial and each dictator (using 'rstanarm', ID set at the random variable; v. 2.18.2; [47]; probability of direction fitted with 'bayestestR'; v. 0.3.0; [48]) to give a visual description of changes in HI and SI scores as trials continued (figure 5).

We calculated the trial where a high (greater than the mean) attribution was made and trial-by-trial changes in attributions when considering pre-existing paranoia (GPTS score). Cumulative link models with multimodal averaging (as with Studies 1 and 2) were used for each dictator. Trial-by-trial analyses between levels of paranoia were visualized separately for harmful intent and self-interest attributions for each dictator (figure 6).

Finally, we ran an exploratory analysis on the combined datasets to establish whether pre-existing paranoia, overall harmful intent attributions, overall self-interest attributions and sex were associated with more unfair decisions made by participants when they took the role of the dictator following being the receiver. It was clear in the task that participants were making decisions for a new partner, as opposed to the partners they had been paired with in the previous trials. Participants made six dictator decisions in total. We used a mixed-effects binomial regression model (using package 'lme4'; v. 1.1.21; [46]) to assess this question, with ID and decision trial (1–6) as random effects. The model

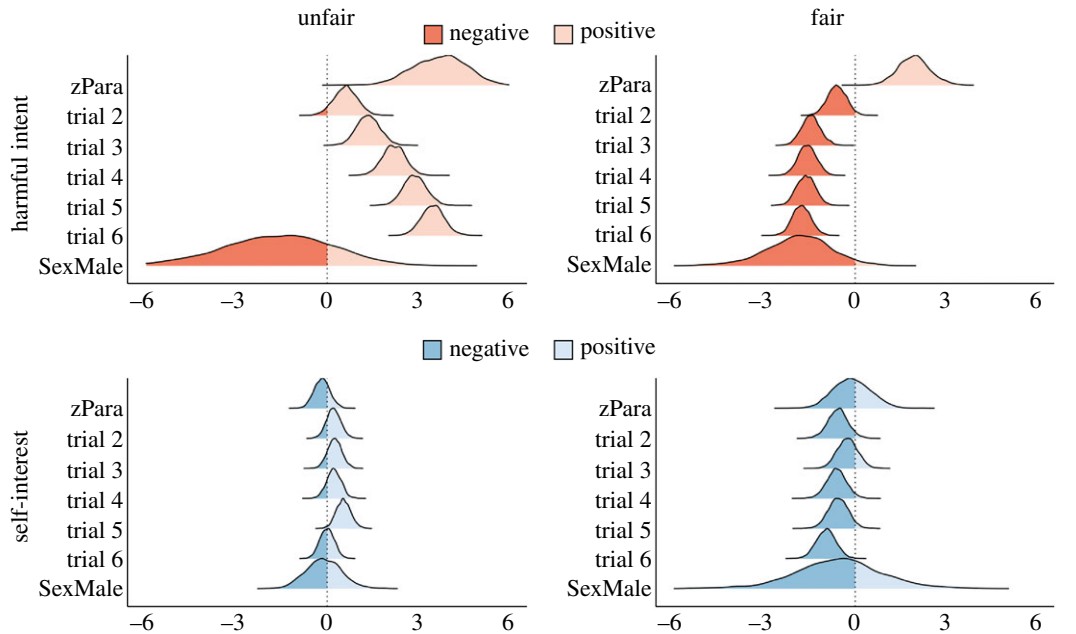

**Figure 5.** Probability distributions of β-coefficient from linear mixed-effects models representing HI and SI attributions of the whole population by unfair and fair dictators between trials 2–6 when compared with trial 1. Probability distributions of β-coefficients modulated by paranoia (zPara; scaled and centred GPTS scores) and being a male (SexMale) when compared with being a female are also included.

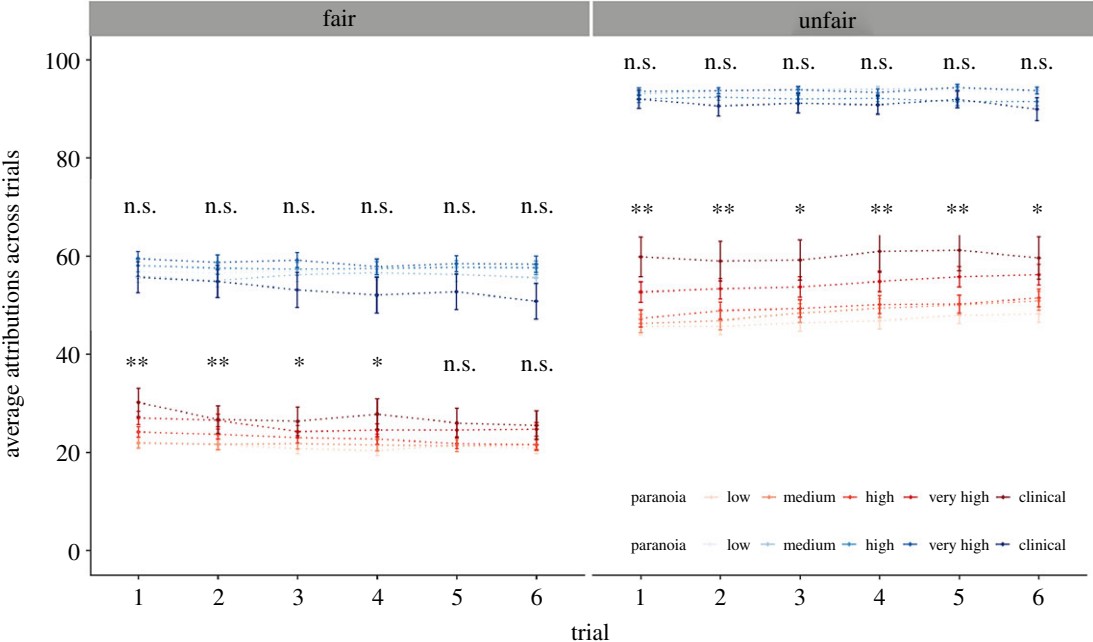

**Figure 6.** Each plot displays mean and s.d. for harmful intent (red) and self-interest (blue) attributions, faceted by dictator, graded in colour by paranoia division. Group comparison significant values represent HI/SI score—paranoia for each trial. $*p < 0.05$, $**p < 0.01$, n.s., not significant.

was unable to converge with both overall harmful intent and self-interest attributions included, so we ran separate models that included each.

## 4.2. Results

See electronic supplementary material, appendix E for the density distributions of scores for each dictator and trial.

### 4.2.1. Order effects

Being initially exposed to a more unfair dictator predicted a decrease in HI attributions for fair (−3.61, 95% CI: −4.38, −2.85) and unfair dictator conditions (−16.70, 95% CI: −19.50, −13.84) in the context of the whole population. Being initially exposed to a more unfair dictator predicted a decrease in self-interest attributions when playing fair (−5.89, 95% CI: −8.05, −3.74) and unfair dictator conditions (−1.66, 95% CI: −2.61, −0.71). Paranoia predicted an increase in HI attributions for both dictators in these models (fair dictator: 1.92, 95% CI: 0.91, 2.94; unfair dictator: 3.47, 95% CI: 1.84, 5.11), but not in SI attributions.

### 4.2.2. Trial-by-trial analysis

See figure 5 (electronic supplementary material, appendix F for confidence intervals) for overall changes in HI and SI scores for each dictator from trials 1–6 across the population.

Paranoia predicted earlier trials in which a high HI score (greater than the mean) was triggered for both unfair (−0.08, 95% CI: −0.14, −0.01) and fair (−0.08, 95% CI: −0.14, −0.02) dictators, although this was not true for SI scores. Additionally, paranoia predicted an overall decrease in scores between the first and the sixth trial for fair (−0.70, 95% CI: −1.54, −0.03) but not unfair dictators, and this was not true for SI scores for either dictator condition (see figure 6 for visual summary).

### 4.2.3. Dictator decision analysis

Paranoia predicted more unfair decisions made by participants (0.66, 95% CI: 0.31, 1.02), as did being male (2.19, 95% CI: 1.44, 2.95) and overall self-interest attributions (0.63, 95% CI: 0.31, 0.96), whereas overall harmful intent attributions (0.27, 95% CI: −0.07, 0.62) and the order in which participants had been partnered with dictators (0.035, 95% CI: −0.25, 0.96) did not affect decisions.

## 5. Discussion

We undertook two studies to test the sensitization model of paranoia using a multi-round dictator game. This controlled experimental design models social inferences about the intentions of a 'dictator' (playing partner) over successive interactions and varying conditions of fair behaviours. In Study 1, we tested the effect of self-reported paranoid beliefs on the attribution of harmful intent. In Study 2, we tested the effect of anxiety, worry and interpersonal sensitivity in modulating these effects.

In line with our predictions, paranoia was associated with earlier and higher levels of harmful intent attribution across all conditions, and higher levels of harmful intent attribution as partners were increasingly unfair in their division of resources. Contrary to predictions, we found no meaningful effects of anxiety, interpersonal sensitivity or worry on the attribution of harmful intent. Self-interest attributions were only modulated by dictator type and the order in which participants were partnered with dictators. An internal meta-analysis highlighted that paranoia was associated with greater reductions of harmful intent attributions in fair dictator conditions over six trials, but not unfair dictators, where harmful intent attributions remained consistent. Additionally, in the meta-analysis regardless of paranoia, harmful intent attributions increased over trials with unfair dictators and decreased over trials with fair dictators. Finally, prior paranoia, but not in-the-moment attributions of harmful intent, predicted more selfish decisions with a new partner in line with prior evidence [29].

Our data provide additional evidence for the sensitization model in paranoia. Our findings converge with previous game theory studies on paranoia that measured attribution of harmful intent using between-subject single-shot designs. In previous studies that used dictator games, paranoia predicted greater harmful intent attributions relative to partner fairness [28,29]. This new study replicated these findings and additionally showed through the use of a within-group design and serial interactions that paranoia was associated with faster and larger attributions of harmful intent relative to partner fairness, suggesting increased sensitivity to perceived threat in interpersonal interactions. This is in line with previous findings from studies using a range of alternative paradigms. Simulated social exclusion with the 'cyberball' game increased state paranoia in non-clinical individuals with high trait paranoia [49], in individuals at high risk of psychosis [50] and patients with paranoid delusions [51]. Experience sampling studies have found that moments of subjective stress [52–54] and physiological arousal [55] predict an increase in paranoia. Similarly, immersion in a stressful social environment, either in virtual reality [54] or a genuine city street [56], increased state paranoia. Additionally, our

results of overall higher self-interest attributions at all levels of paranoia that are only modulated by dictator type are consistent with prior evidence [28–30], demonstrating the specificity of prior paranoia on momentary inferences instead of a general bias in social reasoning.

Our data also converge with theories of social learning. Models of social impression formation in healthy populations suggest that impressions of 'bad' others are more volatile and hence updated more quickly when a putatively bad agent becomes fairer [57]. Our findings that paranoia was associated with higher initial baseline harmful intent attributions, and also greater reductions in harmful intent attributions in fair partner conditions, provide convergent evidence that pre-existing paranoia may both lead to higher baseline impressions of harmful intent and concurrently amplify belief volatility.

Counter to our predictions, we did not find any effect of anxiety or worry on the attribution of harmful intent. Cognitive models of paranoia [58–60] cite worry and anxiety as maintaining paranoid ideation based on a range of prior evidence. Worry has been found to be present at high levels in highly paranoid people [61], and psychological treatment for worry has been shown to reduce paranoia in a targeted randomized controlled trial [62]. Similarly, induction of stress has been shown to increase state paranoia, mediated by anxiety [6,56], in addition to anxiety predicting higher state paranoia in ambiguous virtual environments [63]. Given the strength of prior evidence, we think it unlikely that anxiety and worry play no part in paranoia and suggest two possibilities for why no effect was found in this study. The first may be that we measured harmful intent attributions for specific events and general worry and anxiety may be more involved in maintaining paranoid ideation (i.e. promoting paranoid rumination) than amplifying in-the-moment paranoid attributions. Indeed, currently models of paranoia suggest that anxiety and worry are maintenance factors for paranoid thoughts [16], and worry and experience sampling studies suggest that proximal worry and anxious rumination have a larger effect on paranoia than in time-lagged analyses [64]. Thus, in-the-moment attributions of harmful intent may be more dependent on momentary worry and anxiety prior to social interactions, and it is not certain that traits will become relevant to live social inferences. Secondly, other predisposing factors (e.g. trauma; [16]) not measured may be more relevant to the relationship between general anxiety and harmful intent attributions.

Contrary to our prediction, we found that interpersonal sensitivity was not associated with harmful intent attributions. A recent systematic review reported a strong relationship between interpersonal sensitivity and trait paranoia, but a variable and unclear relationship with state paranoia [65]. For example, using a general population sample, virtual reality studies have found an association between state paranoia and overall interpersonal sensitivity [66], even when adjusting for confounders [67,68]. However, when using 'real world' stooges, an association with state anxiety was only found with the separation anxiety subscale [69]. However, we did not find a positive relationship between harmful intent attributions and any subscale of the interpersonal sensitivity measure when we included them in separate models. Like anxiety and worry, it may be that the influence of trait interpersonal sensitivity on momentary paranoia is dependent on different immediate social circumstance, e.g. when in the presence of another person [66–68], or alternatively that interpersonal sensitivity may only relate to maintaining paranoid thoughts and not momentary harmful intent attributions.

We also note some limitations to this study. As with previous designs, our study used crowd-sourcing platforms. This affords us a much larger, more representative sample than university or community samples [33], with higher response rates [70], greater experimental naivety and larger chances of replication [71], although our data drew solely on a UK population. However, given our exclusion criterion (participants had to fail both control questions to be removed), it is possible that some participants did not respond accurately due to poor attention, potentially leading to inflated effect sizes [33]. We note, however, that previous studies have found online participants to produce equal or better-quality data than laboratory participants for the same task [72]. Additionally, it is not clear to what extent those who score above the clinical mean on the paranoia scale in this study resemble patients with paranoid delusions. Given such a large sample, it would be surprising if at least some of the high scorers did not have delusions, although it is also the case that those most disabled by psychosis may be least able to participate in computer-based studies.

Our game theory paradigm measured harmful attributions in ambiguously motivated, loss-inducing, online interaction. One potential limitation is the extent to which participants were sceptical and believed they were being deceived by the experimenters. We found no relationship between scepticism and harmful intent attributions, and likewise, our findings have replicated previous evidence using a similar manipulation [28,29]. One additional question is the extent to which our findings generalize to diverse social situations. As noted above, the results reported here converge with those reported in

experience sampling studies of everyday interactions and immersive experimental studies, suggesting that they also reflect the operation of common cognitive mechanisms. However, the specific differences in how paranoia manifests in online and offline contexts have yet to be tested and we feel this is something that needs further research.

# 6. Conclusion

We have demonstrated that paranoid ideation leads to quicker and exaggerated attributions of harmful intent, but not attributions of self-interest, in a motivationally ambiguous, live online social task. Our findings support the theory of sensitization in paranoia—specifically, that pre-existing paranoid beliefs reflect a heightened sensitivity to social stress which increases attributions of harmful intent. We also show in a within-group design that the cognitive processes involved in the detection of social threat through fairness are at least partially distinct. The finding that anxiety, interpersonal sensitivity overall and worry did not predict attributions of harmful intent suggests that general anxiety, interpersonal sensitivity as a single measure and worry may mediate paranoid rumination rather than in-the-moment attributions. Future studies will employ game theory paradigms in patient groups to investigate the relationship between clinical and non-clinical paranoia. At a neural level, evidence of the involvement of the mesolimbic dopamine system in psychosis suggests that future studies should investigate how dopamine modulates threat attribution in illness and health.

Ethics. Study 1 was approved by the Kings College London ethics board (MRS-17/18-8312). All data were collected in September 2018 using Prolific Academic (hereafter Prolific; www.prolific.ac), an online crowd-sourcing platform. Study 2 was approved by the Kings College London ethics board (LRS-18/19-9281). Data were collected in February 2019 using Prolific.

Data accessibility. All data and analysis scripts are available at the Open Science Framework: doi:10.17605/OSF.IO/U92RG (https://osf.io/u92rg/).

Authors' contributions. J.M.B. initially devised the studies. J.M.B. constructed the multi-round dictator game. J.M.B. and N.R. revised the multi-round dictator game. J.M.B. collected the data, analysed the data and wrote initially the draft of the manuscript. J.M.B., Q.D., O.R., N.R., V.B. and M.A.M. critically revised the manuscript.

Competing interests. The authors declare that the research was conducted in the absence of any commercial or financial relationships that could be construed as a potential conflict of interest.

Funding. J.M.B. is supported by the UK Medical Research Council (grant no. MR/N013700/1) and King's College London member of the MRC Doctoral Training Partnership in Biomedical Sciences.

Acknowledgement. We thank Uri Hertz for kindly sending his avatar images for use in this game.

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
