## [Reviewer comments · Royal Society Open Science]

Review History

RSOS-191525.R0 (Original submission)

Review form: Reviewer 1 (Hisham Ziauddeen)

Is the manuscript scientifically sound in its present form?

Yes

Are the interpretations and conclusions justified by the results?

Yes

Is the language acceptable?

Yes

Do you have any ethical concerns with this paper?

No

Have you any concerns about statistical analyses in this paper?

No

Recommendation?

Accept with minor revision (please list in comments)

Comments to the Author(s)

This is a lovely paper.

The authors have designed and executed a rigorous set of studies and analyses, all of which have been pre-registered or been very clear where they have deviated from the pre-registration and why. The manuscript is very clear, detailed and well written. The data and scripts have been made available and the overall design and approach are in keeping with the best practices of open science. The discussion is carefully considered and succinct.

I do not have any major comments on the manuscript. I list a few minor comments below that would be helpful to have addressed and I do not think they should trouble the authors unduly:

1. With respect to Study 2, given the strong correlations between the GPTS and the ISM and its subscales, how robust is the cumulative link model to strongly correlated dependent variables? This would be important for considering the additional effects of the ISM subscales over and above the GPTS scores.
2. The authors interpret the decline in harmful attributions in the high paranoia groups in the fair dictator trials as indicating belief volatility. Could this be considered instead as an effect of higher baseline harmful attribution in high paranoia subgroups? i.e. Intentions are harmful unless proved otherwise rather than harmful intentions are more malleable? I appreciate that this may be my reading too much into Figure 6 but given that the effects are significant even accounting for the order effect, perhaps this might be a reasonable interpretation?
3. Please expand STAI and ISM on their first appearance.

Review form: Reviewer 2 (Philip Corlett)

Is the manuscript scientifically sound in its present form?

Yes

Are the interpretations and conclusions justified by the results?

No

Is the language acceptable?

Yes

Do you have any ethical concerns with this paper?

No

Have you any concerns about statistical analyses in this paper?

No

Recommendation?

Major revision is needed (please make suggestions in comments)

Comments to the Author(s)

I read and enjoyed Barnby and colleagues' manuscript, reporting their large online studies of the relationship between paranoia and behavior on the dictator game. They find that people with higher paranoia are more sensitive to the fairness of the dictator's offers, manifest as changes in their attributions of harmful intent at different (increased) rates as paranoia increases.

The report was pre-registered and the manuscript deviates little (and appropriately) from the pre-registered plan.

However, I do feel some features of the data could be unpacked further, should a revision be invited.

For example, can the authors speculate on why the self-interest manipulation was more impactful overall?

Furthermore, it would be helpful to expand the presentation of the developing attributions in figure 2 - for both the fair and unfair dictators, it does not appear that the learning curves are steepest in the very high paranoia groups - though it is difficult to see the data. Isn't this rate of change of intention the point where one might expect the strongest manifestation of the sensitization process? Perhaps I misunderstood, or maybe the 50/50 case confuses things but why aren't the evolving attributions displayed for that condition too?

I realize this is a long and very detailed paper (appropriately), I think the authors ought to consider analyzing and presenting the data from the conditions in which the paranoid subjects proposed offers. Did these always come after the Dictator offers? It would be interesting to know how these participants behaved as a function of their paranoia. More specifically, might there be more 'tit for tat' carryover/correlation between perceived harmful intent from the Dictator to paranoid participants' subsequent proposals. This may not be straightforward to compute given the counterbalancing, but I feel it represents an intriguing possible confirmation of some of the author's hypotheses.

I identify myself as Phil Corlett

Decision letter (RSOS-191525.R0)

02-Jan-2020

Dear Mr Barnby,

The editors assigned to your paper ("Paranoia, sensitisation and social inference: findings from two large-scale, multi-round behavioural experiments.") have now received comments from reviewers. We would like you to revise your paper in accordance with the referee and Associate Editor suggestions which can be found below (not including confidential reports to the Editor). Please note this decision does not guarantee eventual acceptance.

Please submit a copy of your revised paper before 25-Jan-2020. Please note that the revision deadline will expire at 00.00am on this date. If we do not hear from you within this time then it will be assumed that the paper has been withdrawn. In exceptional circumstances, extensions may be possible if agreed with the Editorial Office in advance. We do not allow multiple rounds of revision so we urge you to make every effort to fully address all of the comments at this stage. If deemed necessary by the Editors, your manuscript will be sent back to one or more of the original reviewers for assessment. If the original reviewers are not available, we may invite new reviewers.

When submitting your revised manuscript, you must respond to the comments made by the referees and upload a file "Response to Referees" in "Section 6 - File Upload". Please use this to

document how you have responded to the comments, and the adjustments you have made. In order to expedite the processing of the revised manuscript, please be as specific as possible in your response.

- Data accessibility

If you wish to submit your supporting data or code to Dryad (<http://datadryad.org/>), or modify your current submission to dryad, please use the following link:
<http://datadryad.org/submit?journalID=RSOS&manu=RSOS-191525>

- Competing interests

- Authors' contributions

- Acknowledgements

- Funding statement

on behalf of Dr Simone Schnall (Associate Editor) and Essi Viding (Subject Editor)
openscience@royalsociety.org

Associate Editor's comments (Dr Simone Schnall):

Associate Editor: 1

Comments to the Author:

Dear Mr. Barnby:

I write to you regarding manuscript RSOS-191525 entitled "Paranoia, sensitisation and social inference: findings from two large-scale, multi-round behavioural experiments," which you submitted to Royal Society Open Science.

I have received two evaluations of the manuscript from reviewers who were chosen because of their substantial expertise relating to your research. Before turning to their reviews, I independently read your paper myself.

Both reviewers were overall impressed with your work. While Reviewer 1 was highly enthusiastic and had only minor comments, Reviewer 2 suggested doing further analyses that might shed light on the processes under investigation. I concur with both of them, and therefore invite you to prepare a revision that addresses their concerns.

Both reviewers were very specific, so I refer you to their reports. Regarding the additional analyses suggested by Reviewer 2, please consider to what extent they would be appropriate, and if you chose to pursue them, where they might fit best, either within the main text, or in the supplement.

In addition, please consider the following points:

- 1) You used large samples, which is commendable, but it was not clear how you arrived at sample sizes. It would be useful to have an indication of statistical power, even if it was calculated after the fact.
- 2) You combined data from two studies for further analysis, which makes sense and is informative. However, it is unusual to call this a separate study, such as "Study 3". It would be more appropriate to refer to it as an internal meta-analysis, and describe analyses accordingly.
- 3) There is a bit of a jump between the two studies, and Study 2 is presented without any rationale, or background literature. I suggest adding brief discussion sections for both studies that interpret the results. After the results of Study 1 in particular there needs to be a transition into the logic behind Study 2; right now it seems a bit unconnected. Study 2 therefore needs more of an introduction section.
- 4) The method sections are very long because they also include theory that would be better placed in the introduction. For example, lines 54-65 review the literature on games, which would

fit better much earlier. As a rule of thumb, method sections typically don't have many references because key concepts or theory need to be described already in the introduction.

5) Related to the point above, predictions typically come at the end of introduction sections, after the study rationale has been outlined, so I suggest moving them from the methods sections. Importantly, rather than just listing the predictions (as may be appropriate for a pre-registration document), there also needs to be some justification for *_why_* those particular predictions are sensible, that is, reference should be made to existing theory or other findings. Overall I recommend more clearly separating introduction (=theory) and method (=implementation) sections.

6) Although the manuscript is overall very well-written, some sections, especially parts of the methods and results, read a bit like unconnected bullet points. Please use more of a narrative, incl. pulling separate sentences together into full paragraphs.

7) You present a lot of data, which is great, of course, but given that there is so much information, it would help the reader if you could be as concise as possible. In particular, the legends for tables and figures sometimes repeat the identical information (e.g., Tables 1 and 2). It would be more efficient to refer back to what was already said whenever possible, and only add new information when relevant. Legends also differ regarding whether they are only descriptive of the variables, or also give interpretation of the findings. The latter may not be necessary given that it is already in the text. Also, please arrange variables in the same order across tables (e.g., paranoia, sex). Was there a reason why age was not included in Table 1?

8) Some of the font sizes of your figures are very small. I would urge you to consider what size will still be readable in a journal format. Here it would be worth thinking creatively what kind of information should appear in the same figures, or whether it would be better to break up some of the information to increase readability.

9) The lack of an effect for anxiety and worry is indeed puzzling. Of the three possible explanations, the first suggestion, that it may not apply as much to in-the-moment attributions, would benefit from elaboration, especially since you also list it as the most likely reason in your concluding section. This is an important conceptual point worth bringing out more. In contrast, your third interpretation, namely that the online paradigm was not suitable, does not strike me as plausible. Indeed, you also give reasons why this is unlikely, in which case it would make more sense to drop this suggestion.

Some additional minor issues are:

1) Abstract: You use the word "predicted" five times. I suggest varying the language more.

2) Lines 152-153: I'm assuming all participants were invited to complete the second session, and only a subset did so. Please make this clear here, and later when you describe the follow-up session.

3) Line 154: Add mean and SD for age.

4) Lines 159-161: Sentence could probably be simplified to read: "High trait paranoia will be associated with increased attributions of harmful intent, but not attributions of self-interest." I would encourage you to go through all predictions and describe them in the most concise way.

5) Line: 164: "will" should read "with"

6) Sometimes your use of "all" is redundant. For example, when saying "all data (line 182)", or "all (...) scores" (line 184) it is implied that this would be the case for all data points. Please go through such instances to check whether the word is needed.

- 7) Line 223 (and elsewhere): Use full numbers (i.e., with decimal points).
- 8) Line 236: Here only refer to portion of figure that concerns Study 1.
- 9) Line 254 (and elsewhere): “effecting” should read “affecting”
- 10) Line 528: Omit “(changeable)” since this is implied in “volatile”
- 11) Line 546: Add “another”
- 12) Line 582 (and elsewhere): avoid contractions; for example, instead of “didn’t”, say “did not”

I hope that you take advantage of the time and expertise that the reviewers have contributed to your work. I look forward to possibly receiving a revised manuscript and I would like to thank you for considering RSOS as an outlet for your research.

Sincerely,

Simone Schnall, Ph.D.
Associate Editor, RSOS

Comments to Author:

Reviewers' Comments to Author:

Reviewer: 1

Comments to the Author(s)

This is a lovely paper.

The authors have designed and executed a rigorous set of studies and analyses, all of which have been pre-registered or been very clear where they have deviated from the pre-registration and why. The manuscript is very clear, detailed and well written. The data and scripts have been made available and the overall design and approach are in keeping with the best practices of open science. The discussion is carefully considered and succinct.

I do not have any major comments on the manuscript. I list a few minor comments below that would be helpful to have addressed and I do not think they should trouble the authors unduly:

1. With respect to Study 2, given the strong correlations between the GPTS and the ISM and its subscales, how robust is the cumulative link model to strongly correlated dependent variables? This would be important for considering the additional effects of the ISM subscales over and above the GPTS scores.
2. The authors interpret the decline in harmful attributions in the high paranoia groups in the fair dictator trials as indicating belief volatility. Could this be considered instead as an effect of higher baseline harmful attribution in high paranoia subgroups? i.e. Intentions are harmful unless proved otherwise rather than harmful intentions are more malleable? I appreciate that this may be my reading too much into Figure 6 but given that the effects are significant even accounting for the order effect, perhaps this might be a reasonable interpretation?
3. Please expand STAI and ISM on their first appearance.

Reviewer: 2

Comments to the Author(s)

I read and enjoyed Barnby and colleagues' manuscript, reporting their large online studies of the relationship between paranoia and behavior on the dictator game. They find that people with higher paranoia are more sensitive to the fairness of the dictator's offers, manifest as changes in their attributions of harmful intent at different (increased) rates as paranoia increases.

The report was pre-registered and the manuscript deviates little (and appropriately) from the pre-registered plan.

However, I do feel some features of the data could be unpacked further, should a revision be invited.

For example, can the authors speculate on why the self-interest manipulation was more impactful overall?

Furthermore, it would be helpful to expand the presentation of the developing attributions in figure 2 - for both the fair and unfair dictators, it does not appear that the learning curves are steepest in the very high paranoia groups - though it is difficult to see the data. Isn't this rate of change of intention the point where one might expect the strongest manifestation of the sensitization process? Perhaps I misunderstood, or maybe the 50/50 case confuses things but why aren't the evolving attributions displayed for that condition too?

I realize this is a long and very detailed paper (appropriately), I think the authors ought to consider analyzing and presenting the data from the conditions in which the paranoid subjects proposed offers. Did these always come after the Dictator offers? It would be interesting to know how these participants behaved as a function of their paranoia. More specifically, might there be more 'tit for tat' carryover/correlation between perceived harmful intent from the Dictator to paranoid participants' subsequent proposals. This may not be straightforward to compute given the counterbalancing, but I feel it represents an intriguing possible confirmation of some of the author's hypotheses.

I identify myself as Phil Corlett

Author's Response to Decision Letter for (RSOS-191525.R0)

See Appendix A.

Decision letter (RSOS-191525.R1)

10-Feb-2020

Dear Mr Barnby,

It is a pleasure to accept your manuscript entitled "Paranoia, sensitisation and social inference: findings from two large-scale, multi-round behavioural experiments." in its current form for publication in Royal Society Open Science. The comments of the reviewer(s) who reviewed your manuscript are included at the foot of this letter.

Please ensure that you send to the editorial office an editable version of your accepted

manuscript, and individual files for each figure and table included in your manuscript. You can send these in a zip folder if more convenient. Failure to provide these files may delay the processing of your proof. You may disregard this request if you have already provided these files to the editorial office.

on behalf of Dr Simone Schnall (Associate Editor) and Essi Viding (Subject Editor)
openscience@royalsociety.org

Associate Editor Comments to Author (Dr Simone Schnall):

Dear Mr. Barnby,

Thank you for the thorough revision of the manuscript, and the care you and your colleagues have taken to address the reviewers', and my comments. I am satisfied with the changes, and am happy to accept the paper for publication. It will make a great addition to the literature, so I look forward to seeing it in print.

Sincerely,
Simone Schnall

Appendix A

Dear Dr Schnall,

We would first like to thank the editor and the reviewers for their time in constructing a critique of our work. The comments pay close attention to detail and provide excellent insight into areas we may have missed. The manuscript is now stronger for it.

Please find the response to all of the comments below - we have addressed each individually.

We look forward to your response.

Yours sincerely,

Joseph Barnby

Editor Comments:

Thank you to the editor for her detailed, informative, and constructive comments on the manuscript. We have addressed each independently and highlighted changes in the text where appropriate in yellow.

1) You used large samples, which is commendable, but it was not clear how you arrived at sample sizes. It would be useful to have an indication of statistical power, even if it was calculated after the fact.

A power analysis has been included simulating a multiple regression model with at least 7 predictors to detect an effect size of 0.1 (lines 116 and 277). We have justified our use of more participants than the power analysis required.

2) You combined data from two studies for further analysis, which makes sense and is informative. However, it is unusual to call this a separate study, such as "Study 3". It would be more appropriate to refer to it as an internal meta-analysis and describe analyses accordingly.

We have adjusted the name of 'Study 3' to 'Internal Meta-Analysis'.

3) There is a bit of a jump between the two studies, and Study 2 is presented without any rationale, or background literature. I suggest adding brief discussion sections for both studies that interpret the results. After the results of Study 1 in particular there needs to be a transition into the logic behind Study 2; right now it seems a bit unconnected. Study 2 therefore needs more of an introduction section.

We have summarised our first study and altered the text introducing study 2. We feel this provides a more connected transition between studies. We have included our specific predictions in the introductory sections for each study. Because of the strong replication in the core findings from each study, we have opted to retain a single general discussion at the end.

4) The method sections are very long because they also include theory that would be better placed in the introduction. For example, lines 54-65 review the literature on games, which would fit better much earlier. As a rule of thumb, method sections typically don't have many

references because key concepts or theory need to be described already in the introduction.

The methods sections are fairly lengthy as they provide detailed structure to a complex analysis method, while including sufficient detail to allow the reader to understand our decisions. Thus, we include a number of references for replicability and transparency, including specific packages in R that are necessary to complete the analyses, and the detailed methodological decisions made. All refer to justification in our computing and statistical approach that we believe best placed in the methods section.

We wanted to also ensure transparency and replicability of our serial dictator game paradigm and provides specific details about the game in the methods but introduce the general principles of the game and its prior use in the introduction (line 55-66).

5) Related to the point above, predictions typically come at the end of introduction sections, after the study rationale has been outlined, so I suggest moving them from the methods sections. Importantly, rather than just listing the predictions (as may be appropriate for a pre-registration document), there also needs to be some justification for *_why_* those particular predictions are sensible, that is, reference should be made to existing theory or other findings. Overall I recommend more clearly separating introduction (=theory) and method (=implementation) sections.

Predictions have been moved from the methods section to the introductory part of each study in the manuscript and rationale have been described for their inclusion.

6) Although the manuscript is overall very well-written, some sections, especially parts of the methods and results, read a bit like unconnected bullet points. Please use more of a narrative, incl. pulling separate sentences together into full paragraphs.

We have edited the text to form full paragraphs and avoid 'bullet point' style writing.

7) You present a lot of data, which is great, of course, but given that there is so much information, it would help the reader if you could be as concise as possible. In particular, the legends for tables and figures sometimes repeat the identical information (e.g., Tables 1 and 2). It would be more efficient to refer back to what was already said whenever possible, and only add new information when relevant. Legends also differ regarding whether they are only descriptive of the variables, or also give interpretation of the findings. The latter may not be necessary given that it is already in the text. Also, please arrange variables in the same

order across tables (e.g., paranoia, sex). Was there a reason why age was not included in Table 1?

Duplicate information has been removed from the main text where possible, and instead the main text summarizes the result and refers to the specific statistics in the main table when available. If a table doesn't contain the statistic, the statistic is still stated in the text. Legends have been homogenized to remove any interpretation of the findings. Variables are now listed in tables in the same order.

Age is not included in table 1 because it was not a component of the final top model in the averaging process (deemed by the package MuMIn) and thus wasn't included in the statistical output in R. While age is included in Study 2, age is a very weak predictor of both harmful intent and self-interest attributions.

8) Some of the font sizes of your figures are very small. I would urge you to consider what size will still be readable in a journal format. Here it would be worth thinking creatively what kind of information should appear in the same figures, or whether it would be better to break up some of the information to increase readability.

We have edited the figures to make them more readable and make them as succinct as possible, including homogenising and enlarging the text values for each axis and legend.

9) The lack of an effect for anxiety and worry is indeed puzzling. Of the three possible explanations, the first suggestion, that it may not apply as much to in-the-moment attributions, would benefit from elaboration, especially since you also list it as the most likely reason in your concluding section. This is an important conceptual point worth bringing out more. In contrast, your third interpretation, namely that the online paradigm was not suitable, does not strike me as plausible. Indeed, you also give reasons why this is unlikely, in which case it would make more sense to drop this suggestion.

We have expanded upon the point in question (line 576 - 581). We have also dropped the final suggestion that the paradigm was not suitable.

Minor issues:

1) Abstract: You use the word "predicted" five times. I suggest varying the language more.

We have edited the abstract to vary the language.

2) Lines 152-153: I'm assuming all participants were invited to complete the second session, and only a subset did so. Please make this clear here, and later when you describe the follow-up session.

We have adjusted the wording to make it clear some people did not take part in the second session despite being invited.

3) Line 154: Add mean and SD for age.

As age was included in the first study as an ordinal set of answers (e.g. 18-24, 24-30), I am unable to provide the mean and SD for age here.

4) Lines 159-161: Sentence could probably be simplified to read: "High trait paranoia will be associated with increased attributions of harmful intent, but not attributions of self-interest." I would encourage you to go through all predictions and describe them in the most concise way.

We have moved and reworded each specific prediction from the methods to the introduction at the start of each study (see points 3 and 5).

5) Line: 164: "will" should read "with"

This has been changed.

6) Sometimes your use of "all" is redundant. For example, when saying "all data (line 182)", or "all (...) scores" (line 184) it is implied that this would be the case for all data points. Please go through such instances to check whether the word is needed.

I have revised and adjusted sentence where appropriate.

7) Line 223 (and elsewhere): Use full numbers (i.e., with decimal points).

This has been adjusted in the text.

8) Line 236: Here only refer to portion of figure that concerns Study 1.

This has been adjusted in the text.

9) Line 254 (and elsewhere): “effecting” should read “affecting”

This has been adjusted in the text.

10) Line 528: Omit “(changeable)” since this is implied in “volatile”

This had been adjusted in the text.

11) Line 546: Add “another”

This has been adjusted in the text

12) Line 582 (and elsewhere): avoid contractions; for example, instead of “didn’t”, say “did not”

This has been adjusted in the text.

Reviewer 1 Comments

Thank you to Reviewer One for their constructive comments and their appraisal of the paper. We have addressed each comment independently and highlighted changes in the text in blue.

1) With respect to Study 2, given the strong correlations between the GPTS and the ISM and its subscales, how robust is the cumulative link model to strongly correlated dependent variables? This would be important for considering the additional effects of the ISM subscales over and above the GPTS scores.

Thank you very much for this consideration. This has been something we have revisited in light of your comments and have addressed empirically in the manuscript.

We ran the models again to double check our results in case of collinearity by fitting the residuals of the ISM to take into account the variance explained by paranoia and found that the effect of total interpersonal sensitivity with paranoia in the model [Harmful Intent ~ ISM (residuals) + GPTS + Dictator + Age + Sex + (1|ID)] remained (beta = -0.29). Additionally, running variance inflation factor (vif) analysis using the “car” package in R which suggested low collinearity (vif < 2) for each predictor. This suggests that the cumulative link mixed models were able to robustly deal with correlated predictors. Additionally, we ran commonality analyses that suggested that paranoia made up 98% of the variance, the ISM made up 28% of the variance, and together they made up -26% of the variance, suggesting that paranoia has a suppressing effect on the ISM, exaggerating its relationship with attributions.

Therefore, we also explored models that included the ISM and its subscales as predictors in absence of paranoia [attribution ~ predictor + age + sex + Dictator + (1|ID)]. These models suggested an absence of any effects that were previously reported, consistent with correlations in figure 3. We have noted this exploratory addition in the methods for transparency. This may indicate the dependence of interpersonal sensitivity toward its contribution to prior paranoia, leading to changes harmful intent attributions, but not on its own.

We have updated table 3 to report all estimates of predictors in their own independent models [attribution ~ predictor + age + sex + Dictator + (1|ID), which is stated in the methods] and created Appendix D to include our previous analyses that included all subscales of the interpersonal sensitivity measure in one model, and also included paranoia

in all models as we had preregistered. We also report our preregistered statistical analyses in the text, but reserve the table presented to reader with the models that were absent of paranoia.

The direction of effect of interpersonal sensitivity or its subscales wasn't a key feature in the discussion or in our conclusions, but we have updated the description where it featured in the discussion to highlight this deviation. We have also edited the abstract to correct for this.

2) The authors interpret the decline in harmful attributions in the high paranoia groups in the fair dictator trials as indicating belief volatility. Could this be considered instead as an effect of higher baseline harmful attribution in high paranoia subgroups? i.e. Intentions are harmful unless proved otherwise rather than harmful intentions are more malleable? I appreciate that this may be my reading too much into Figure 6 but given that the effects are significant even accounting for the order effect, perhaps this might be a reasonable interpretation?

While it is correct that paranoia leads to a higher baseline in harmful intent attributions (we have made this clearer in line 560-564) this doesn't rule out greater belief volatility. Thus, we are interpreting the results as both higher baseline attributions of harmful intent *and* greater belief volatility (as there is a sharper decline in scores, and not just sustained attributions). Additionally, neither the fair nor unfair scores are close to the floor, and the decline in harmful intent attributions is only significant for the fair condition which have a lower baseline; therefore, it isn't just baseline dependent. We have made this clearer in the discussion.

3) Please expand STAI and ISM on their first appearance.

This has been adjusted in the text.

Review 2 Comments

Thank you to Professor Corlett for his constructive comments and his appraisal of the manuscript. We have completed his suggested analysis to the extent we were able to and addressed each of his comments independently and highlighted changes in the text in green.

1) For example, can the authors speculate on why the self-interest manipulation was more impactful overall?

We have included a section in the discussion (lines 528-530 and 552-556) that highlight this result and speculates on its meaning with regard with prior work.

2) Furthermore, it would be helpful to expand the presentation of the developing attributions in figure 2 - for both the fair and unfair dictators, it does not appear that the learning curves are steepest in the very high paranoia groups - though it is difficult to see the data. Isn't this rate of change of intention the point where one might expect the strongest manifestation of the sensitization process? Perhaps I misunderstood, or maybe the 50/50 case confuses things but why aren't the evolving attributions displayed for that condition too?

The presentation of the developing attributions for unfair and fair dictators are better described in Figure 6 in the internal meta-analysis, where the change in attribution over time is clearer, and also statistically described through the information theoretic analysis in the same section that show paranoia leads to reduced scores over time in fair but not unfair conditions, where it remains constant.

The sensitisation hypothesis in this context suggests that paranoia will lead to heightened responses to social stimuli. We indeed find that paranoia influences early harmful intent attributions in fair conditions. However, following repeated administration it's been unclear whether this would experimentally lead to increases or decreases over time. While we find sensitisation generally for the whole population, we in fact find the opposite when taking into account paranoia – paranoia reduced scores in fair conditions but had no effect on scores over time in unfair conditions. We do mention these conclusions in the internal meta-analysis.

The 50/50 case has been omitted from figure 2 and 6 as all the trials were presented randomly, so trying to visually describe it with a graph would be somewhat misleading to what the actual data suggests (as it would artificially order trials for the sake of visuals).

3) I realize this is a long and very detailed paper (appropriately), I think the authors ought to consider analysing and presenting the data from the conditions in which the paranoid subjects proposed offers. Did these always come after the Dictator offers? It would be interesting to know how these participants behaved as a function of their paranoia. More specifically, might there be more 'tit for tat' carryover/correlation between perceived harmful intent from the Dictator to paranoid participants' subsequent proposals. This may not be straightforward to compute given the counterbalancing, but I feel it represents an intriguing possible confirmation of some of the author's hypotheses.

Each participant made their dictator decisions following all decisions made by their three consecutive partners. It was made clear in the task instructions that they would be making decisions for a new participant, not one they had already been partnered with.

We have however included a brief analysis that aims to assess relevant factors associated with dictator decisions, an exploratory logistic mixed effects regression model with multi-model averaging with both harmful intent and self-interest ($\text{DecisionValue} \sim [\text{Harmful Intent Attributions}/\text{Self Interest Attributions}] + \text{Paranoia} + \text{Sex} + (1|\text{ID}) + (1|\text{Decision})$) has been completed and added to the end of the internal meta-analysis and a sentence stating it's replication of prior studies has been added in the discussion (line 525).